# ELASTIC-MDM: EFFICIENT MASKED DIFFUSION MODELS WITH VARIABLE SEQUENCE LENGTHS

## ABSTRACT

Discrete masked diffusion models (MDMs) enable parallel denoising with bidirectional context but incur unnecessary compute by encoding the entire sequence at every step and by assuming a fixed output length. We propose the *Removed–Masked Diffusion Model* (Elastic-MDM), which redesigns the state space by making a REMOVED token absorbing and excluding it from the Transformer input; a single reverse pass per step couples token denoising with a lightweight gap–count head that predicts how many removed tokens to (re)activate between consecutive unmasked tokens, enabling variable-length decoding. We derive a model-aligned objective without timestep weights and train with schedule randomization. Empirically, Elastic-MDM delivers substantial wall-time savings at similar quality on benchmark datasets, closely tracks the training length distribution without preset caps, and improves structured (JSON) synthesis. This shows that Elastic-MDM offers a simple, practical path to efficient, variable-length discrete diffusion.

## 1 INTRODUCTION

Autoregressive (AR) language models have achieved remarkable success in text generation by factorizing the data distribution into a product of conditional next-token distributions and decoding left-to-right (Achiam et al., 2023; Clusmann et al., 2023; Zhao et al., 2023). However, this unidirectional one-token-at-a-time generation limits parallelization and complex tasks such as planning and reasoning requiring bidirectional context understanding (Yang et al., 2023). Masked diffusion models (MDMs) offer a compelling alternative: it casts generation as iterative denoising from a fully noised token sequence (Austin et al., 2021; Hoogeboom et al., 2021; Campbell et al., 2022). This paradigm allows many positions to update in parallel and enables global-context reasoning (Gong et al., 2022; Li et al., 2022).

Despite these advantages, standard MDMs suffer from two practical limitations that hinder broader adoption. **(i) Inefficiency**. At each denoising step, the model encodes the entire sequence, even though only a subset of masked tokens are updated. This is *unnecessary* compute on masked tokens that remain fixed. This encoding design inflates both training and inference time. **(ii) Fixed length**. Most discrete diffusion models operate with a preset sequence lengths, which constrains their ability to adapt to variable-length outputs when the sequence length is not known in advance. A broad body of concurrent work underscores how central these bottlenecks are—exploring masked/blocked attention on the compute side, and semi-autoregressive generation, insertion/deletion predictors as well as length predictor on the length side (Sahoo et al., 2025; Arriola et al., 2025; Zhang et al., 2025; Kim et al., 2025). Yet most approaches address *only one side at a time*—either compute or length—while largely preserving the conventional MDM state space and training objective.

To address these limitations, we introduce Elastic Masked Diffusion Model (Elastic-MDM), a masked diffusion framework designed for both *efficiency* and *variable-length generation*. Elastic-MDM augments the standard state space with an absorbing [REMOVED] token: once a token transitions to [REMOVED], a token is excluded in the subsequent sequence, so the forward trajectory monotonically shortens. We mirror this transition to define the reverse process with a denoiser and a gap count predictor. At each reverse step, the input contains only already–unmasked tokens together with scheduler-selected [MASK] tokens. Then, a denoiser predicts the contents of the current [MASK] tokens, while a lightweight *gap-count predictor* estimates, for every pair of consecutive

unmasked tokens, how many `[REMOVED]` tokens should lie; the scheduler then insert that many masks according to the noise schedule. We further provide a training recipe tailored to this process, combining a single-pass denoising loss with a categorical gap-count loss and schedule-aware noise injection. By construction, the model encodes only *active* positions, eliminating redundant computation, and adapts sequence length on the fly by predicting and instantiating the number of reactivated tokens per gap.

Through our experiments, we show that Elastic-MDM delivers substantial speedups in wall-clock time and FLOPs while reaching comparable perplexity compared to standard MDMs. Beyond speed, we observe that Elastic-MDM better aligns the training and generation length distributions and proves advantages in structured-output settings such as JSON generation thanks to the variable-length generation capability of Elastic-MDM.

Our contributions are: (i) **State-space redesign** for masked diffusion—mask as transient and a removed absorbing state—so persistently hidden tokens are never encoded; (ii) **Model-aligned training objective** from NELBO that yields a single-pass denoiser loss and a categorical gap–count loss; (iii) **Variable-length generation** via *gap–count prediction* between unmasked tokens; (iv) **Training & inference efficiency** — active token-only encoding cuts computation cost in both training and inference; analysis and measurements agree across schedulers.

## 2 RELATED WORKS

**Discrete diffusion for text.** Early discrete diffusion works focus on a general corruption–denoising Markov chains in categorical spaces and provide the comprehensive training methods which can be applied to various applications (Austin et al., 2021; Hoogeboom et al., 2021). For text, several works such as DiffusionBERT (He et al., 2022) and SEDD Lou et al. (2023) instantiate this framework with a mask-as-absorbing process over token vocabularies. Subsequent works simplify the training framework and make the masked discrete diffusion more practical: MDLM Shi et al. (2024) and MD4 (Shi et al., 2024) propose a standard training objective function based on the simple weighted cross-entropy loss with clean parametrization. More recently, scaling studies demonstrate feasibility at larger scale, *e.g.,* DiffusionLLaMA (Gong et al., 2024), LLaDA (Nie et al., 2025), DREAM 7B (Ye et al., 2025), displaying noticeable improvement in performance competitive to the recent autoregressive large language models.

**Boosting efficiency in discrete diffusion.** Recent work accelerates discrete diffusion models by reducing either the *per-step* cost or the number of *denoising* steps. One line of work adapt KV-caching to bidirectional denoising, to lower the cost of each step (Ma et al., 2025; Hu et al., 2025). Another line exploits the parallel nature of discrete diffusion to shrink the denoising iteration: for example, confidence-based decoding Yu et al. (2025) and adaptive parallel decoding Israel et al. (2025) adjusts how many masked token to reveal per step. These directions are complementary to ours. Rather than caching or merely widening parallel decoding, we *change the state space*: a removed token becomes the absorbing state and is excluded it from the Transformer input, and we introduce a variable-length mechanism which *predicts* how many removed tokens to (re)activate—cutting compute from both sides (fewer active positions per step and fewer steps under suitable schedulers). Concurrently, EsoLM (Sahoo et al., 2025) also excludes tokens not scheduled to be unmasked from the attention, but it assumes a fixed-length generation whereas our model supports variable-length generation.

**Variable-length generation beyond fixed-length.** Early attempts achieved variable length by operating in an edit space—explicit insert/delete or tag$\rightarrow$ action—so the length of generated sequence can be variable during decoding Johnson et al. (2021); Reid et al. (2022). Subsequently, semi-autoregressive diffusion models sequentially decode a block of tokens in the diffusion manner using the previous blocks as the condition (Arriola et al., 2025), where they still are limited with a fixed-length of block generation. More recently, several works bring variable length inside MDMs: DDOT jointly denoises token values and positions to insert/shift spans (Zhang et al., 2025); DAEDAL and DreamOn use diffusion-LM variants to control span length during infilling (Li et al., 2025) and (Wu et al.); FlexMDM gradually inserts mask tokens and unmask (Kim et al., 2025). These are close in spirit to ours in enabling variable-length generation, but they remain in the original diffusion state

space and do not address computation efficiency gained by omitting masked tokens that will not be unmasked.

## 3 PRELIMINARIES

Discrete denoising diffusion models (Austin et al., 2021; Hoogeboom et al., 2021) define a Markov corruption-denoising process over categorical tokens via transition matrices. A *masked diffusion model* (MDM) (Sahoo et al., 2024; Shi et al., 2024) is a widely used special case of discrete diffusion for language modeling in which corruption moves toward an absorbing mask state ([MASK]), and, once reached, it remains fixed.

**Notation**. Let $\mathcal{V}$ be a vocabulary of size $K$ and extend it with the mask to $\tilde{\mathcal{V}} = \mathcal{V} \cup \{[\text{MASK}]\}$ of size $K+1$. A single token $x$ denotes an element of $\tilde{\mathcal{V}}$; we write $\mathbf{m} \in \tilde{\mathcal{V}}$ for the mask token. A clean sequence of length $L$ is $\mathbf{x}_0 = (x_0^{(1)}, \ldots, x_0^{(L)})$ with $x_0^{(l)} \in \mathcal{V}$ for all $l$. Diffusion timestep is indexed by $t \in \{0, 1, \ldots, T\}$ with per-step survival rate $\alpha_t \in [0, 1]$ that decreases monotonically, $\alpha_0 = 1$ and $\alpha_T = 0$.

**Forward process**. In the forward diffusion, the MDM gradually corrupts the clean sequence $\mathbf{x}_0$ by stochastically replacing original tokens with the mask, converging to $\mathbf{x}_T = (\mathbf{m}, \ldots, \mathbf{m})$. The token for the position $l \in \{1, \ldots, L\}$ is independently corrupted according to

$$q\left(x_t^{(l)} \mid x_{t-1}^{(l)}\right) = \begin{cases} \delta_{\mathbf{m}}, & x_{t-1}^{(l)} = \mathbf{m}, \\ \frac{\alpha_t}{\alpha_{t-1}} \cdot \delta_{x_{t-1}^{(l)}} + \left(1 - \frac{\alpha_t}{\alpha_{t-1}}\right) \cdot \delta_{\mathbf{m}}, & x_{t-1}^{(l)} \in \mathcal{V}, \end{cases} \tag{1}$$

where $\delta_y$ is the point mass at token $y \in \tilde{\mathcal{V}}$.

**Reverse process and training.** The reverse distribution $p_\theta(\mathbf{x}_{t-1}|\mathbf{x}_t)$ is parameterized by a denoising diffusion model and trained with the Negative ELBO (NELBO) (Sohl-Dickstein et al., 2015):

$$\mathcal{L}(\theta) = \mathbb{E}_{\mathbf{x}_0, q}\left[-\log p_\theta(\mathbf{x}_0|\mathbf{x}_1) + \sum_{t=2}^{T} D_{\text{KL}}\big(q(\mathbf{x}_{t-1}|\mathbf{x}_t, \mathbf{x}_0) \,\big\|\, p_\theta(\mathbf{x}_{t-1}|\mathbf{x}_t)\big) + D_{\text{KL}}\big(q(\mathbf{x}_T|\mathbf{x}_0) \,\big\|\, p_\theta(\mathbf{x}_T)\big)\right]. \tag{2}$$

Rather than optimizing the full NELBO in Eq 2, most masked diffusion models (*e.g.*, MDLM (Sahoo et al., 2024)) use a repameterized objective: instead of learning $p_\theta(\mathbf{x}_{t-1}|\mathbf{x}_t)$ directly, they set $p_\theta(\mathbf{x}_{t-1}|\mathbf{x}_t) = q(\mathbf{x}_{t-1}|\mathbf{x}_t, \mathbf{x}_0 = \hat{\mathbf{x}}_\theta(\mathbf{x}_t))$, *i.e.*, the forward posterior conditioned on a model-predicted clean sequence $\hat{\mathbf{x}}_\theta(\mathbf{x}_t)$. Combined with variance-reduction techniques such as Rao–Blackwellization (Sahoo et al., 2022) yields the simplified training loss:

$$\mathcal{L}(\theta) = \mathbb{E}_{t, q}\left[\frac{\alpha_{t-1} - \alpha_t}{1 - \alpha_t} \sum_{l : \mathbf{x}_t^l = \mathbf{m}} \log f_\theta(\mathbf{x}_t)^{(l)}(x_0^{(l)})\right], \tag{3}$$

in which $f_\theta(\mathbf{x}_t)^{(l)}(v)$ is the probability assigned to token $v$ in the softmax output from the model $\theta$.

**Generation.** Sampling in a standard masked diffusion model starts from the fully masked sequence $\mathbf{x}_T = (\mathbf{m}, \ldots, \mathbf{m})$ of *fixed length $L$*. For timesteps $t = T, T-1, \ldots, 1$, a noise scheduler selects a subset $\mathcal{S}_t \subseteq \{l : x_t^{(l)} = \mathbf{m}\}$ to reveal. The denoiser then produces token distributions only at the selected masked positions, $\{f_\theta(\mathbf{x}_t)^{(l)}\}_{l \in \mathcal{S}_t}$ over $\mathcal{V}$, while unmasked tokens are copied through. The chosen positions are filled (*e.g.*, by sampling or argmax), yielding $\mathbf{x}_{t-1}$ while other masked positions remain masked. This iterative unmasking continues until no masks remain, producing $\mathbf{x}_0 \in \mathcal{V}^L$.

**Limitations of standard masked diffusion**. Typical MDMs suffer from two key inefficiencies. First, at each denoising step the model encodes the entire length-$L$ sequence, even though the update only act on $\mathcal{S}_t$. Masked tokens that will not be revealed at the current step contribute little beyond a constant mask embedding and positional markers, yet they still participate fully in the model's encoding. Second, the model assumes a fixed sequence length throughout training and sampling. In practice many tasks require outputs shorter than this preset $L$; the fixed-length generation forces the model to generate full-length sequences (or pad to $L$), which is unnecessarily costly when shorter outputs would suffice. These issues motivate the efficiency-oriented redesign introduced next.

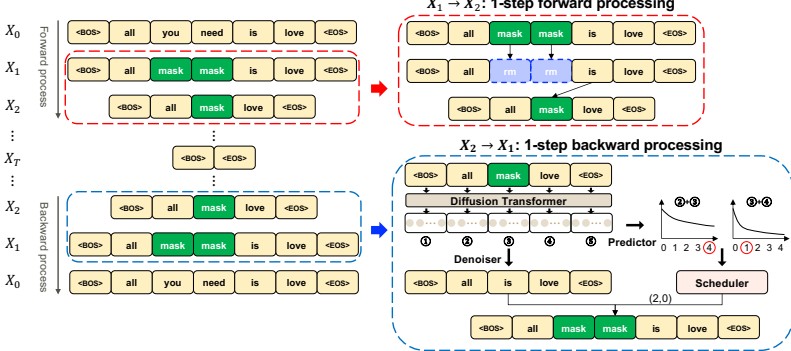

Figure 1: **Elastic-MDM overview.** Forward: clean $\rightarrow$ m(transient) $\rightarrow$ **r** (absorbing), removing **r**. Reverse: a single pass denoises masks ($p_\theta$) and predicts gap counts ($p_\phi$); the scheduler activates sampled masks to form $\mathbf{x}_{t-1}$—enabling variable-length outputs and lower compute cost.

## 4 ELASTIC MASKED DIFFUSION MODEL

We propose a novel masked diffusion framework, dubbed Elastic Masked Diffusion Model (Elastic-MDM), that augments the state space with a new token, [REMOVED] (**r**). In our design, **r** is the *absorbing* state, while the standard mask **m** serves only as a *transient* state on the way from a clean token to the absorbing state (clean $\rightarrow$ **m** $\rightarrow$ **r**). Once a token becomes **r**, we remove it from the current token sequence, so the sequence length decreases over the forward trajectory. Building on this forward process, we parameterize a denoising diffusion model to mirror the forward. In specific, at each reverse step the input sequence contains only already–unmasked tokens and a scheduler-selected masked tokens at each reverse process while all **r** tokens are absent. The diffusion model then unmasks the masked tokens, predicts how many **r** tokens should lie for every pair of consecutive unmasked tokens, and, based on these predicted counts, we sample how many mask tokens to activate for the next step according to the noise scheduler. The Elastic-MDM forward and backward processes are illustrated in Fig. 1. This design eliminates unnecessary processing of masked tokens and enable variable-length generation,

### 4.1 UPDATED SETUP WITH A REMOVED TOKEN

Let the extended vocabulary be $\hat{\mathcal{V}} = \mathcal{V} \cup \{\mathbf{m}, \mathbf{r}\}$, where **m** is the (transient) mask and **r** is the new absorbing *removed* token. We assume boundary tokens <BOS>, <EOS> $\in \mathcal{V}$ that are fixed in the entire diffusion process. At diffusion step $t$, we write the token sequence as

$$\mathbf{x}_t = (\texttt{<BOS>}, x_t^{(1)}, \ldots, x_t^{(L_t)}, \texttt{<EOS>}), \qquad x_t^{(i)} \in \mathcal{V}, \tag{4}$$

where $L_t$ is the number of *unmasked* interior tokens. Since positions that enter **r** are removed from the sequence, $L_t$ is nonincreasing in $t$, and $x_T = (\texttt{<BOS>}, \texttt{<EOS>})$. We also introduce the vector of *gap counts*

$$\boldsymbol{\ell}_t = (\ell_t^{(0)}, \ell_t^{(1)}, \ldots, \ell_t^{(L_{t+1})}) \in \mathbb{Z}_{\geq 0}^{L_{t+1}+1}, \tag{5}$$

where $\ell_t^{(i)}$ is the number of inactive positions—tokens in $\{\mathbf{m}, \mathbf{r}\}$—that lie between the $i$-th and $(i{+}1)$-th unmasked tokens in $\mathbf{x}_t$. Note that the number of $\boldsymbol{\ell}_t$ is $L_{t+1} + 1$ since the gaps counted at time $t$ between unmasked tokens are those between the unmasked and newly converted masked tokens of $\mathbf{x}_{t-1}$.

The state at time $t$ is the pair $\mathbf{z}_t = (\mathbf{x}_t, \boldsymbol{\ell}_t)$.

### 4.2 FORWARD PROCESS

Under Elastic-MDM, the corruption proceeds in two adjacent-step phases: clean vocabulary tokens may be *masked* at rate $1 - \alpha_t/\alpha_{t-1}$, and any token that was masked in the previous step becomes

*removed* (absorbing) in the next step. For a single token $x \in \hat{\mathcal{V}}$, the forward transition probability is

$$q(x_t|x_{t-1}) = \begin{cases} \delta_{\mathbf{r}}, & x_{t-1} \in \{\mathbf{m}, \mathbf{r}\}, \\ \frac{\alpha_t}{\alpha_{t-1}}\delta_{x_{t-1}} + \frac{\alpha_{t-1}-\alpha_t}{\alpha_{t-1}}\delta_{\mathbf{m}}, & x_{t-1} = x_0. \end{cases} \tag{6}$$

Given the single-token forward kernel in Eq. 6, the sequence-level transition $\mathbf{z}_{t-1} = (\mathbf{x}_{t-1}, \boldsymbol{\ell}_{t-1}) \rightarrow \mathbf{z}_t = (\mathbf{x}_t, \boldsymbol{\ell}_t)$ is induced directly by applying the kernel tokenwise and updating the gap counts accordingly; we provide the full derivation in Appendix B.1.

## 4.3 REVERSE PROCESS

To mirror the forward dynamics, we parametrize and factorize the reverse transition at step $t$ from $\mathbf{z}_t = (\mathbf{x}_t, \boldsymbol{\ell}_t)$ to $\mathbf{z}_{t-1} = (\mathbf{x}_{t-1}, \boldsymbol{\ell}_{t-1})$ as

$$p_{\theta,\phi}(\mathbf{x}_{t-1}, \boldsymbol{\ell}_{t-1} \mid \mathbf{x}_t, \boldsymbol{\ell}_t) \ = \ p_\phi(\boldsymbol{\ell}_{t-1} \mid \mathbf{x}_t, \boldsymbol{\ell}_t)\, p_\theta(\mathbf{x}_{t-1} \mid \mathbf{x}_t, \boldsymbol{\ell}_t, \boldsymbol{\ell}_{t-1}). \tag{7}$$

where $p_\theta$ is the denoiser realized by the diffusion network, and $p_\phi$ is the *gap count* predictor implemented as a lightweight neural network attached on top of the diffusion model. With this factorization, our reverse proceeds in two designed steps: (1) the gap count predictor estimates how many inactive tokens lie between each pair of unmasked tokens (gap counts), and (2) then the denoiser updates only the masked tokens in the sequence $\mathbf{x}_t$.

**Token denoising ($p_\theta$).** The denoiser outputs a distribution over $\mathcal{V}$ for a masked tokens with the position $k$ in $\mathbf{x}_t$, $f_\theta(\mathbf{x}_t)^{(k)} \in \Delta^K$, and we set

$$p_\theta\big(x_{t-1}^{(k)} \mid \mathbf{x}_t, \boldsymbol{\ell}_t, \boldsymbol{\ell}_{t-1}\big) \ = \ f_\theta(\mathbf{x}_t)^{(k)} \tag{8}$$

while already–unmasked tokens are copied deterministically: $x_{t-1}^{(k)} = x_t^{(k)}$ if $x_t^{(k)} \in \mathcal{V}$. In practice we observed no measurable improvement from conditioning on $_{t,t-1}$; hence we condition on $\mathbf{x}_t$ only.

**Gap count prediction ($p_\phi$).** Let $h_t^{(k)}$ be the diffusion model's final-layer features at the $k$-th token $x_t^{(k)}$. We design the predictor to output a categorical distribution over the number of removed tokens for a pair of adjacent unmasked tokens. Concretely, given $h_t^{(k)}$ and $h_t^{(k+1)}$, the predictor outputs softmax values over discrete counts $0{:}R_{\max}$:

$$p_\phi(\ell_{t-1}^{(k)} \mid \mathbf{x}_t, \boldsymbol{\ell}_t) \ = \ \text{softmax}\, g_\phi\big(h_t^{(k)}, h_t^{(k+1)}\big) \in \Delta^{R_{\max}},$$

in which $R_{\max}$ is the maximum possible number of inactive tokens in our model, which coincides with the global maximum sequence length. For the same reason above, we condition on $\mathbf{x}_t$ only.

## 4.4 ELASTIC-MDM TRAINING OBJECTIVE

Applying the NELBO (Eq. 2) to our formulation with the parameterized reverse distribution over $\mathbf{z}_t = (\mathbf{x}_t, \boldsymbol{\ell}_t)$ yields the following objective

$$\mathcal{L}_{Elastic-MDM} = \sum_{t=2}^{T} \mathbb{E}_{\mathbf{z}_0, q(\mathbf{z}_t|\mathbf{z}_0)}\Big[ D_{\mathrm{KL}}\big(q(\mathbf{z}_{t-1}|\mathbf{z}_t, \mathbf{z}_0) \,\|\, p_{\theta,\phi}(\mathbf{z}_{t-1}, \mid \mathbf{z}_t))\big)\Big]. \tag{9}$$

Using the factorization in Sec. 4.3 and the exact forward posterior $q(\mathbf{z}_{t-1}|\mathbf{z}_t, \mathbf{z}_0)$, the KL in Eq. 9 reduces to the sum of two negative log-likelihoods for token denoising and gap count prediction:

$$\mathcal{L}_{Elastic-MDM} = -\sum_{t=2}^{T} \mathbb{E}_{\mathbf{z}_0, q(\mathbf{z}_t|\mathbf{z}_0), q(\mathbf{z}_{t-1}|\mathbf{z}_t, \mathbf{z}_0)}\Bigg[ \underbrace{\sum_{x_t^{(i)}=\mathbf{m}} \log p_\theta(x_0^{(i)}|\mathbf{x}_t, \boldsymbol{\ell}_t, \boldsymbol{\ell}_{t-1})}_{\text{denoiser loss}} + \underbrace{\sum_i^{L_t} \log p_\phi(\ell_{t-1}^{(i)}|\mathbf{x}_t, \boldsymbol{\ell}_t)}_{\text{gap count prediction loss}}\Bigg]. \tag{10}$$

Full derivation is deferred to Appendix B.3. We simultaneously compute each term in Eq. 10 with a single forward pass of the diffusion network on $\mathbf{x}_t$ and jointly optimize them at every step. Regarding gradient routing, we use gradients from the gap count loss to update both the diffusion backbone and the predictor head, whereas the denoiser loss updates only the backbone.

---

**Algorithm 1** RMDM Sampling with Variable Length

---

**Require:** noise schedules $\{\alpha_t\}_{t=0}^T$; models $f_\theta$ (token denoiser) and $g_\phi$ (gap–count head)

1: $\mathbf{x}_T \leftarrow \langle \texttt{<BOS>}, \texttt{<EOS>} \rangle$

2: **for** $t = T, \ldots, 1$ **do**

3:     extract $h_t^{(0)}, h_t^{(1)}, \ldots, h_t^{(L_t)}, h_t^{(L_t+1)}$ from the diffusion model's backbone and $\mathbf{x}_t$

4:     **(gap prediction)** for $k = 0, \ldots, L_t - 1$:
    $\pi_t^{(k)} \leftarrow \mathrm{softmax}(g_\phi(h_t^{(k)}, h_t^{(k+1)}))$; sample $\tilde{\ell}_t^k \sim \mathrm{Cat}(\pi_t^{(k)})$

5:     **(single forward)** for all masked token with the position $l$,
    sample $x_{t-1}^{(l)} \sim p_\theta(\cdot \mid \mathbf{x}_t)$; copy already-unmasked tokens through to obtain $\mathbf{x}_{t-1}$

6:     $p_{\mathrm{mask}}(t) \leftarrow \frac{\alpha_{t-2} - \alpha_{t-1}}{1 - \alpha_{t-1}}$

7:     **(activate masks)** for $k = 0, \ldots, L_t$:
    $L_t^{(k)} \sim \mathrm{Binomial}(\ell_t^{(k)}, p_{\mathrm{mask}}(t))$;
    insert $L_t^k$ masks between $x_t^{(k)}$ and $x_t^{(k+1)}$ to form $\mathbf{x}_{t-1}$

8: **end for**

9: **return** $\mathbf{x}_0$

---

**Noise-schedule randomization during training.** Unlike reparameterized MDM objectives that admit a $T \to \infty$ limit and a continuous–time view, our loss in Eq. 10 depends explicitly on the finite horizon $T$. Hence we cannot recover a continuous–time objective by sending $T$ to infinity. To avoid overfitting to a single discretization and to retain robustness to the scheduler and the number of denoising steps used at inference, we adopt a practical mixture strategy: at every update we randomize the schedule. Concretely,

$$t \sim U\{1, \ldots, T\}, \qquad 1 - \alpha_t \sim U[\beta, \omega], \quad 0 \le \beta < \omega \le 1,$$

*i.e.*, we sample the step and draw a clipped mask rate within $[\beta, \omega]$. This exposes the model to a spectrum of discretizations and corruption levels during training, yielding better generalization to unseen schedulers and timestep budgets at test time.

*Remark* (Comparison to the standard MDM loss). Relative to the usual reparameterized MDM objective as in Eq. 3, our objective differs in three ways: (i) **No timestep weights** — once $t$ is sampled, there is no explicit schedule weight in the denoiser loss (ii) **Progressive supervision** — we supervise the immediate reverse transition by predicting $\mathbf{x}_{t-1}$ in the denoiser loss, rather than predicting $\mathbf{x}_0$ directly from $\mathbf{x}_0$. (iii) **Additional gap-count loss** — beyond token denoising, we add a categorical likelihood term for the gap count predictor, which learns how many removed tokens should appear between adjacent unmasked tokens.These differences are not ad hoc; they follow directly from our reverse parameterization with an absorbing removed state, so Eq. 10 is a well-aligned objective for our diffusion modeling.

## 4.5 Variable-length generation using Elastic-MDM

Starting from the fully noised state $x_T = (\texttt{<BOS>}, \texttt{<EOS>})$, we iterate for $t = T, \ldots, 1$ with a single forward pass per step:

1. **Gap counts.** For each pair of consecutive unmasked tokens in $\mathbf{x}_t$, the head $g_\phi$ outputs a categorical distribution $\pi_t^{(k)}$ over counts $r \in \{0, \ldots, R_{\max}\}$; we stochastically sample $\tilde{r}_t^{(k)} \sim \mathrm{Cat}(\pi_t^{(k)})$.

2. **Token denoising.** We decode all masked positions with $p_\theta(\cdot \mid \mathbf{x}_t, \boldsymbol{\ell}_{t-1})$ and copy unmasked tokens.

3. **Mask activation.** Given the scheduler's activation rate, we activate a random subset of the predicted removed tokens, $\tilde{r}_t^{(k)}$, and insert those $\mathbf{m}$ between the two unmasked tokens to form the next sequence $\mathbf{x}_{t-1}$.

Because removed tokens are never fed to the model and newly inserted masks depend on the predicted gap counts, the effective length $L_t$ evolves during sampling; consequently, $x_0$ is generated with a *variable length* rather than a pre-fixed length. The full procedure is summarized in Alg. 1.

Table 1: **Inference computation cost of our method under different scheduling strategies.** $T$ is the number of denoising step; $L$ is the target sequence length. We report the total attention compute up to constants. Standard MDM costs $TL^2$ regardless of the scheduler.

| Scheduler | $\alpha_t$ (schedule) | Computation cost | Asymptotic order |
|---|---|---|---|
| Cosine | $\cos\frac{\pi t}{2}$ | $L^2 \sum_{t=1}^{T} \cos^2 \frac{\pi t}{2}$ | $\mathcal{O}(TL^2)$ |
| Linear | $1 - t$ | $L^2 \frac{(2T+1)(T+1)}{6T}$ | $\mathcal{O}(TL^2)$ |
| Exponential | $e^{-\sigma t}$ | $L^2 \frac{1 - e^{-2\sigma T}}{e^{2\sigma} - 1}$ | $\mathcal{O}(L^2)$ |

**Computational efficiency of Elastic-MDM compared to standard MDMs**    Our sampler is *computationally efficient* because removed tokens $\mathbf{r}$ are never fed to the Transformer: attention is computed only over the active sequence at each step. This yields a smaller total attention cost than standard MDM. For the concrete comparison, let $T$ be the number of denoising steps and $L$ the target sequence length. While Elastic-MDM generates variable-length sequences, we assume for analysis that Elastic-MDM generates $L$ tokens as the final output, and the gap count predictor and the denoiser act correctly so that the unmasked token length equals $\alpha_t L$. Then, the exact savings depend on the mask schedule $\{\alpha_t\}$ (Table 1). With the widely used linear schedule, our complexity remains $\mathcal{O}(TL^2)$ but the dominant $TL^2$ term's constant drops to $\frac{1}{3}$, giving substantial speedups, especially for long sequences. With an exponential schedule, the complexity further reduces to $\mathcal{O}(L^2)$, asymptotically improving over the standard $\mathcal{O}(TL^2)$. This is because using the exponential schedule, the initial decoding phases run on much shorter inputs and later phases generate more tokens at a time, which is more efficient when encoding active tokens only like Elastic-MDM. We empirically observe matching *training* and *inference* speedups (Sec. 5.1 and 5.3).

## 5 EXPERIMENTS

**Experimental Setup.**    We evaluate Elastic-MDM on two corpora and sequence budgets: LM1B (Chelba et al., 2013) with $L_{\max} = 128$ and OpenWebText (OWT) (Gokaslan et al., 2019) with $L_{\max}=1024$. Unless noted otherwise, models share the same backbone size and training recipe across methods. We report (i) generative perplexity (Sahoo et al., 2024) scored by a frozen LM evaluator (Qwen-3 (Team, 2025)) on decoded samples, and (ii) wall-clock efficiency (per-sequence latency; batch throughput). Our baselines is standard MDM model, MDLM (Sahoo et al., 2024), for fixed-length diffusion. All baselines use the same tokenizer, maximum length cap, and scheduler family as Elastic-MDM. Complete data, model, optimization scheme, and hardware details are provided in Appendix C.

### 5.1 PERPLEXITY VS. WALL TIME

We sweep the number of denoising steps $T$ and compare quality–speed trade-offs for Elastic-MDM, MDLM, and SEDD. We used the linear scheduler and measure (i) generative PPL (lower is better) and (ii) per-sequence latency on identical hardware.

From the Fig. 2, *Elastic-MDM* consistently attains lower per-sequence latency than MDLM at the same $T$, across all tested denoising budgets $T$. The gap grows with $T$: as more steps are taken, active-only encoding amortize the cost advantage, yielding progressively larger end-to-end speedups. In terms of quality, the generative PPL of Elastic-MDM is on par with—and in several settings modestly better than MDLM. Overall, the quality–speed Pareto curve shifts favorably for Elastic-MDM.

### 5.2 LENGTH-DISTRIBUTION ALIGNMENT

We test whether Elastic-MDM reproduces the empirical training-length distribution at generation time without fixing a global $L$. For each dataset we plot the histogram of target lengths in the

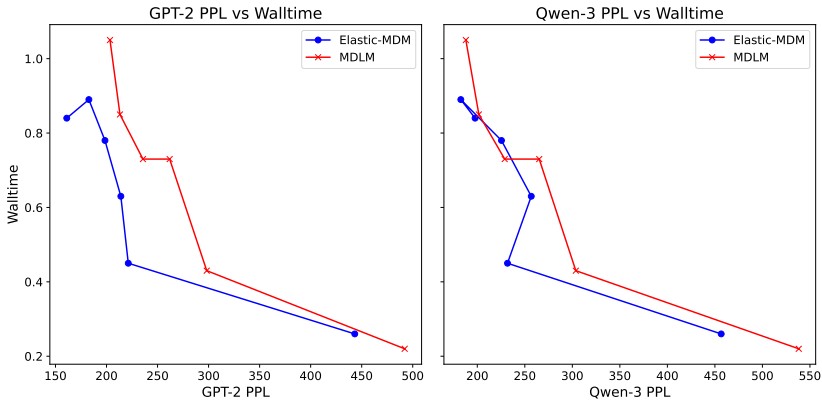

Figure 2: **Quality–speed trade-off on LM1B** ($L$=128). Generative perplexity (Qwen-3 scorer; lower is better) vs. wall time (ms / sample) as $T$ varies. Circles: Elastic-MDM; squares: MDLM.

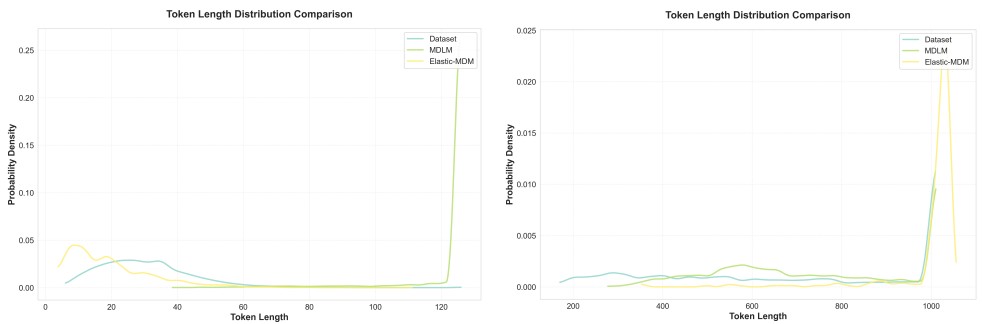

Figure 3: **Length-distribution matching.** Train vs. generated length histograms on LM1B ($L$=128) and OWT ($L$=1024). Elastic-MDM adapts lengths on the fly and follows the training distribution closely; MDLM collapses to a fixed cap.

training set and the histogram of generated lengths from Elastic-MDM (variable) and MDLM (fixed $L$ or padded).

With MDLM, lengths concentrate at the preset cap (128 for LM1B, 1024 for OWT) or at padded values, so the generated-length histogram fails to match the data distribution. In contrast, Elastic-MDM closely follows the training-length histogram: the gap–count predictor reliably allocates variable numbers of tokens between structure anchors.

### 5.3 RUNTIME VS. SCHEDULER FAMILY

We measure per-sequence wall time for three schedules with the same backbone and step budget $T$: linear, p-linear with $p$=2, and exponential. As predicted by our cost model ($\propto L^2 \sum_t \alpha_t^2$), **exponential is fastest**, **p-linear (2-linear) is second**, and **linear is slowest**; see Fig. 4.

### 5.4 STRUCTURED OUTPUT: JSON GENERATION

Schema–constrained JSON generation is naturally variable length: each key–value field may expand to a different number of tokens, and many fields (e.g., arrays, free-form strings) have unbounded length. With a fixed-length denoising process, one must reserve a global budget and either pad or truncate, which wastes computation and often hurts structural validity. In contrast, given Elastic-MDM can exploit JSON structures. Namely, we first instantiate a template (skeleton) that pins all structural tokens (braces, brackets, commas, colons, quotes, and keys). At each reverse step the denoiser fills only the masked *content* slots, while the gap–count head predicts how many additional tokens should appear *between* two anchored structure tokens. This lets the model elastically allocate length per field and per step, something standard fixed-length MDMs cannot do.

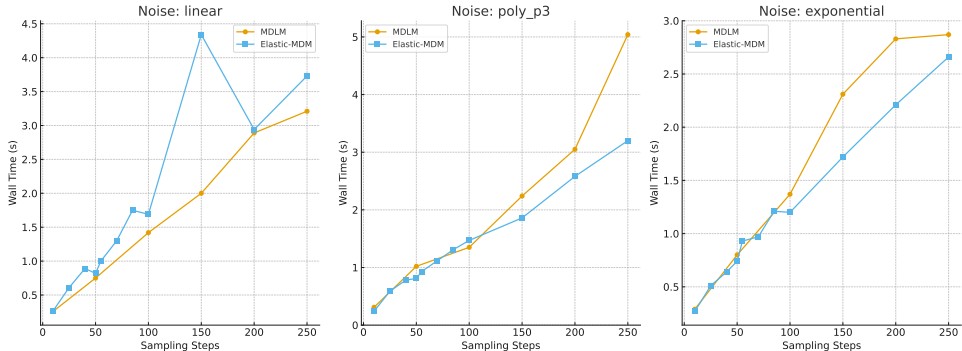

Figure 4: **Scheduler-dependent runtime on OWT** ($L$=1024**, fixed** $T$**).** Measured wall time for Elastic-MDM vs. MDLM under linear/polynomial/exponential schedules. Consistent with analysis, exponential yields asymptotically $O(L^2)$ behavior for Elastic-MDM.

**Task setup and training design.** We evaluate Elastic-MDM on JSON synthesis under predefined schemas. Following the SchemaBench benchmark (Lu et al., 2025), we adopt the schema-only setting, which tests structural fidelity without reasoning. In this task, the model is given a schema and must generate JSON content that conforms to it, validated using the `jsonschema` library (Berman & contributors, 2025). We filter SchemaBench samples to length $\leq$ 1024 tokens, yielding 7,537 instances (6,783 train / 754 validation). As mentioned earlier, JSON generation can be formulated as an infilling task, where the model reconstructs missing content given a schema and structural tokens (e.g., `{ }, [ ], , , :`). Thus, we train our model to infill the missing content with variable-length, where as a baseline, MDLM, is trained using fixed placed holders with a sufficient token size, which are filled with a padding token for remained positions.

**Results.** Our method yields a substantial improvement over MDLM in this setting. On the held-out validation set, MDLM attains only 3.58% valid JSON under the supplied schema, whereas Elastic-MDM reaches 24.01% (Table 2). We score an output as correct if it can be parsed and validated by `jsonschema`. Note that SchemaBench–style JSON generation is typically evaluated with large language models; here we intentionally use a compact backbone to study efficiency and controllability. While this keeps absolute accuracy modest, the *relative* gains from variable-length generation are clear, and we expect accuracy to improve substantially with larger backbones—our state-space redesign and gap prediction are orthogonal to model size.

| Method | Valid (%) |
|---|---|
| MDLM | 3.58% |
| Elastic-MDM | 24.01% |

Table 2: **Validation accuracy on schema-only JSON generation**.

## 6 CONCLUSION

We revisited masked diffusion through the lens of efficiency and flexibility. By introducing a RE-MOVED absorbing token and excluding it from the model input, Elastic-MDM avoids repeatedly encoding persistently hidden sites. A single-pass reverse step then couples token denoising with a gap-count predictor that decides how many removed tokens lie between unmasked tokens, enabling variable-length decoding. Building on this modeling, we derive the objective aligned with the reverse parameterization. Empirically, Elastic-MDM reduces wall time while maintaining comparable generative perplexity. It also matches training length distributions and improves schema-constrained JSON generation. Overall, Elastic-MDM shows that a minimal change to the discrete diffusion state space yields a practical decoder that is lean, parallel, and naturally variable-length.

## ETHICS STATEMENT

We acknowledge the importance of transparency and responsible use of publicly available resources in our research. All datasets, benchmarks, and models employed in this study will be publicly

released for academic research and will be used in full compliance with their respective licenses and intended purposes. No proprietary or private data were used and this research does not involve human subjects or sensitive personal information. We acknowledge that LLMs were used as writing assistants to improve grammar, clarity, and readability of the manuscript.

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

## A  APPENDIX OVERVIEW

This appendix contains:
**Sec. B** – proofs and derivations, including the sequence-level forward transition and the reduction of the NELBO to our training objective in Eqs. 2–10.
**Sec. C** – experimental settings sufficient for reproduction: datasets, architecture and training details, hyperparameters, schedules, and compute.
**Sec. D** – mask schedules, presenting different mask schedules (cosine, linear, exponential, power-linear, clipped) that govern the rate at which tokens are revealed during training.

## B  PROOFS AND DERIVATIONS

The function $g_0$ is a deterministic operation that computes the count vector $\ell_t$ from current sequence $x_t$ and initial state $z_0$. Likewise, the function $g_1$ is a deterministic operation that computes the count vector $\ell_t$ from current sequence $x_t$ and previous state $z_{t-1}$.

### B.1  SEQUENCE-LEVEL TRANSITION $\mathbf{z}_{t-1} \rightarrow \mathbf{z}_t$

Given anchors in $x_t$ and gap counts $\ell_t$, reconstruct the full canvas by interleaving inactive sites; then show how $(x_{t-1}, \ell_{t-1})$ deterministically maps to $(x_t, \ell_t)$ under the forward. *Details and edge cases (BOS/EOS) to be inserted.*

$$q(\mathbf{z}_t|\mathbf{z}_{t-1}) = q(\mathbf{x}_t|\mathbf{z}_{t-1})\, q(l_t|\mathbf{x}_t, \mathbf{z}_{t-1}) \tag{11}$$

$$= \begin{cases} q(\mathbf{x}_t|\mathbf{x}_{t-1}), & \text{if } l_t = g_1(\mathbf{x}_t, \mathbf{z}_{t-1}) \\ 0, & \text{otherwise} \end{cases} \tag{12}$$

$q(\mathbf{x}_t|\mathbf{x}_{t-1})$ is calculated via the single-token factorization.

### B.2  SEQUENCE-LEVEL POSTERIOR DISTRIBUTION

We derive the exact reverse posterior under our forward process. Using Bayes' rule together with this constraint, the posterior factorizes as

$$q(\mathbf{x}_{t-1}, \boldsymbol{\ell}_{t-1} \,|\, \mathbf{z}_t, \mathbf{z}_0) = q(\mathbf{x}_{t-1} \,|\, \mathbf{z}_t, \boldsymbol{\ell}_{t-1}, \mathbf{z}_0) \times q(\boldsymbol{\ell}_{t-1} \,|\, \mathbf{z}_t, \mathbf{z}_0) \,. \tag{13}$$

Then, by definition of $g_0()$, we have

$$q(\mathbf{x}_{t-1} \,|\, \mathbf{z}_t, \boldsymbol{\ell}_{t-1}, \mathbf{z}_0) = \begin{cases} q(\mathbf{x}_{t-1} \,|\, \mathbf{x}_t, \boldsymbol{\ell}_{t-1}, \mathbf{x}_0) & \text{if } \boldsymbol{\ell}_{t-1} = g_0(\mathbf{x}_{t-1}, \mathbf{z}_0) \\ 0, & \text{otherwise.} \end{cases} \tag{14}$$

The posterior is factorized:

$$q(\mathbf{x}_{t-1} \,|\, \mathbf{x}_t, \boldsymbol{\ell}_{t-1}, \mathbf{x}_0) = \Pi_{\mathbf{x}_{t-1}^i = \mathbf{r}} \frac{1-\alpha_{t-2}}{1-\alpha_{t-1}} \Pi_{\mathbf{x}_{t-1}^i = \mathbf{m}} \frac{\alpha_{t-2} - \alpha_{t-1}}{1-\alpha_{t-1}} \tag{15}$$

in which,

$$q\left(\mathbf{x}_{t-1}^i \,\middle|\, \mathbf{x}_t^{i'}, \boldsymbol{\ell}_{t-1}, \mathbf{x}_0\right) = \begin{cases} 1 & \mathbf{x}_t^{i'} \notin \{\mathbf{m}, \mathbf{r}\} \\ 1 & \mathbf{x}_t^{i'} = m \\ \frac{\alpha_{t-2} - \alpha_{t-1}}{1-\alpha_{t-1}} & \mathbf{x}_t^{i'} = r. \end{cases} \tag{16}$$

We also have the posterior of $\ell_t$ by the forward transition definition:

$$q(\boldsymbol{\ell}_{t-1} \,|\, \mathbf{z}_t, \mathbf{z}_0) = \Pi \binom{\boldsymbol{\ell}_t^i}{n_i(\mathbf{x}_t, \boldsymbol{\ell}_t^i)} \tag{17}$$

,where $n_i(\mathbf{x}_t, \boldsymbol{\ell}_t^i)$ indicates number of masked token in $l_t^i$ which can be extracted from $\mathbf{x}_t$ and $\boldsymbol{\ell}_t^i$.

### B.3 FROM NELBO TO THE RMDM OBJECTIVE

**Sketch.** Starting from $\mathcal{L}_{\text{RMDM}} = \mathbb{E}_{z_0,\, q(z_t | z_0)}\big[\text{KL}\big(q(z_{t-1} \mid z_t, z_0) \,\|\, p_{\theta,\phi}(z_{t-1} \mid z_t)\big)\big]$, apply the factorization $p_{\theta,\phi}(z_{t-1} \mid z_t) = p_\phi(\ell_{t-1} \mid x_t)\, p_\theta(x_{t-1} \mid x_t, \ell_{t-1})$, expand the KL, and use the exact forward posterior to separate (i) token values at activated masks and (ii) gap-count terms, yielding Eq. equation **??**. *Full derivation to be inserted.*

**Proof.** Let define the NELBO as following.

$$\mathcal{L}_{\text{NELBO}} = \mathbb{E}_{q(\mathbf{x}_0)}\Big[\sum_{t=1}^{T} \mathcal{L}_{t-1} \,+\, \mathcal{L}_T\Big] \tag{18}$$

$$\tag{19}$$

Then we have:

$$\mathcal{L}_T = \mathbb{E}_{q(\mathbf{x}_0)}\big[\text{D}_{\text{KL}}(q(\mathbf{z}_T \mid \mathbf{z}_0) \,\|\, p(\mathbf{z}_T))\big] = 0.$$

And we also compute $\mathcal{L}_{t-1}$

$$\mathcal{L}_{t-1} = \mathbb{E}_{q(\mathbf{z}_t | \mathbf{z}_0)}\Big[\text{D}_{\text{KL}}\big(q(\mathbf{z}_{t-1} \mid \mathbf{z}_t, \mathbf{x}_0) \,\|\, p_\theta(\mathbf{z}_{t-1} \mid \mathbf{x}_t, \ell_l)\big)\Big] \tag{20}$$

$$= \mathbb{E}_{q(\mathbf{z}_t | \mathbf{z}_0)}\mathbb{E}_{q(\mathbf{z}_{t-1})}\Big[\log \frac{q(\mathbf{x}_{t-1}|\mathbf{z}_t, \mathbf{z}_0, \ell_{t-1})q(\ell_{t-1}|\mathbf{z}_t, \mathbf{z}_0)}{p_\theta(\mathbf{x}_{t-1}|\mathbf{z}_t, \ell_{t-1})p_\theta(\ell_{t-1}|\mathbf{z}_t)}\Big] \tag{21}$$

$$= \mathcal{L}_{t-1} = \mathbb{E}_{q(\mathbf{z}_t | \mathbf{z}_0)}\mathbb{E}_{q(\mathbf{z}_{t-1})}\Big[\log \frac{q(\mathbf{x}_{t-1}|\mathbf{z}_t, \mathbf{z}_0, \ell_{t-1})}{p_\theta(\mathbf{x}_{t-1}|\mathbf{z}_t, \ell_{t-1})}\Big] + \mathbb{E}_{q(\ell_{t-1})}\Big[\log \frac{q(\ell_{t-1}|\mathbf{z}_t, \mathbf{z}_0)}{p_\theta(\ell_{t-1}|\mathbf{z}_t)}\Big]. \tag{22}$$

The first term is computed as follows:

$$\mathbb{E}_{q(\mathbf{z}_{t-1})}\Big[\log \frac{q(\mathbf{x}_{t-1}|\mathbf{z}_t, \mathbf{z}_0, \ell_{t-1})}{p_\theta(\mathbf{x}_{t-1}|\mathbf{z}_t, \ell_{t-1})}\Big] \tag{23}$$

$$= \mathbb{E}_{q(\mathbf{z}_{t-1})}\Big[\log \frac{q(\mathbf{x}_{t-1}|\mathbf{x}_t, \mathbf{z}_0, \ell_{t-1})}{p_\theta(\mathbf{x}_{t-1}|\mathbf{x}_t, \ell_{t-1})}\Big] \tag{24}$$

$$= \mathbb{E}_{q(\mathbf{z}_{t-1})} \sum_{\mathbf{x}_t^{(i)}=\mathbf{m}} -\log p_\theta(\mathbf{x}_0^{(i)} \mid \mathbf{x}_t, \ell_{t-1}) + C. \tag{25}$$

Regarding the second term, we have:

$$\mathbb{E}_{q(\mathbf{z}_{t-1})}\Big[\log \frac{q(\ell_{t-1}|\mathbf{z}_t, \mathbf{z}_0)}{p_\theta(\ell_{t-1}|\mathbf{z}_t)}\Big] = \mathbb{E}_{q(\mathbf{z}_{t-1})} \sum_{i}^{N_t} -\log p_\phi(\mathbb{1}(\hat{\ell}_{t-1}^{(i)} = \ell_{t-1}^{(i)}) \mid \mathbf{z}_t) + C. \tag{26}$$

## C EXPERIMENTAL SETTINGS

### C.1 DATASETS AND PREPROCESSING

For language modeling experiments, Our model is evaluated primarily on the One Billion Words dataset (LM1B)(Chelba et al., 2013) and OpenWebText (OWT)(Gokaslan et al., 2019). LM1B is tokenized with the bert-base-uncased tokenizer at a fixed context length of 128, while OWT uses the GPT-2 tokenizer with sequences wrapped to length 1024 and explicit eos tokens inserted between concatenated documents. Vocabulary size is defined by the categorical space of tokens, with the last index reserved for a special [MASK] token. In LM1B, sequences are truncated or padded to length 128, whereas in OWT the first and last tokens of each batch are forced to eos. Filtering rules are not explicitly detailed, except that LM1B follows prior work in detokenization. For evaluation, LM1B reports perplexity on its standard test split, while OWT—lacking a predefined split—uses the last 100K documents as a held-out validation set.

## C.2 ARCHITECTURE

The backbone model follows a diffusion transformer architecture (Peebles & Xie, 2023). Our configuration consists of 12 layers with hidden size $d_{\text{model}} = 768$ and 12 attention heads, resulting in approximately 110M parameters. We adopt rotary positional embeddings (RoPE; (Su et al., 2024)) with BOS anchoring to stabilize generation, and support a maximum sequence length of 1024 tokens. In addition, we incorporate a regressor head $g_\phi$ that predicts the number of tokens between two observed positions. Timestep conditioning is applied to produce conditioning vectors used for AdaLN modulation. To efficiently handle variable-length sequences, we employ FlashAttention (Dao et al., 2022) with the varlen kernel, enabling memory- and compute-efficient attention over dynamically padded batches.

## C.3 TRAINING HYPERPARAMETERS

We train our models with the AdamW optimizer (learning rate $3 \times 10^{-4}$, $\beta_1 = 0.9$, $\beta_2 = 0.999$, $\epsilon = 10^{-8}$, weight decay 0) and apply gradient clipping at 1.0. A constant warmup learning rate schedule with 2,500 warmup steps is used by default, although alternative schedules such as cosine decay with warmup are also supported. The diffusion horizon is configured as either $T = 0$ (continuous-time) or a fixed-step setting (e.g., $T = 1000$), depending on the experiment. Multiple noise schedules, including log-linear and geometric variants, are implemented with additional randomization strategies (e.g., probabilistic masking or insertion), while theoretical parameters such as $1 - \alpha_t \sim U[\beta, \omega]$, $R_{\max}$, or label clipping to $R_{\max}$ do not appear as explicit configuration options but are instead reflected implicitly in the diffusion loss formulations.

## C.4 COMPUTE RESOURCES

Experiments were conducted on 4×NVIDIA H100 NVL GPUs (94 GB each) with bfloat16 mixed precision. Training was managed with PyTorch 2.2.1, using CUDA 12.4, and key libraries including Transformers 4.53.2 and flash-attn 2.5.6.

# D MASK SCHEDULES

We parameterize corruption by a *survival rate* schedule $\{\alpha_t\}_{t=0}^T \subset [0, 1]$ that is monotonically decreasing with $\alpha_0 = 1$ and $\alpha_T \approx 0$. Let $\bar{t} = t/T \in [0, 1]$. The instantaneous mask rate is $1 - \alpha_t$; the *activation fraction* used in our sampler is $\rho_t := 1 - \alpha_t/\alpha_{t-1}$ (i.e., the fraction of inactive positions that become active between $t-1$ and $t$).

**Cosine.** The cosine schedule gradually preserves more mass early and releases it later:

$$\alpha_t = \cos\left(\frac{\pi}{2}\bar{t}\right), \qquad t = 0, \dots, T.$$

(Optionally use the squared form $\cos^2(\cdot)$(Han et al., 2022); we adopt the unsquared version so that $\alpha_T = 0$ exactly.)

**Linear.** Mass decreases at a constant rate:

$$\alpha_t = 1 - \bar{t}, \qquad t = 0, \dots, T.$$

**Exponential.** Early steps change little while later steps release many tokens at once:

$$\alpha_t = e^{-\sigma\bar{t}}, \qquad \sigma > 0.$$

We set $\sigma = \log\frac{1}{\varepsilon}$ so that $\alpha_T = \varepsilon$ (e.g., $\varepsilon = 10^{-3}$). *Intuition:* because $\alpha_t$ stays high initially and drops sharply near the end, only a few positions are active early while many are activated in later steps—precisely where our active-only encoding yields the largest speedups.

**Power-Linear (p-linear).** A one-parameter family that interpolates between convex/concave reveals:

$$\alpha_t = (1 - \bar{t})^p, \qquad p > 0.$$

Here $p = 1$ recovers linear; $p > 1$ (convex) keeps more mass early and releases it later (closer to exponential), whereas $0 < p < 1$ reveals more mass early (closer to cosine). We use $p$ to sweep how "bursty" the schedule is.

**(Optional) Clipped variants.** Following *clipped* schedules, one may sample a mask rate $1 - \alpha_t \sim$ Uniform$[\beta, \omega]$ with $0 \le \beta < \omega \le 1$ and keep $\alpha_t$ piecewise-linear within the active range; outside $[\beta, \omega]$ we set $\alpha_t \approx 1$ (before) and $\alpha_t \approx 0$ (after). This preserves low-variance gradients when sampling discrete steps.

