# OpenReview forum: "Elastic-MDM: Efficient Masked Diffusion Models with Variable Sequence Lengths"
_ICLR.cc/2026/Conference — Submitted to ICLR 2026_

### Official Review · Reviewer_XDPV · 2025-10-23

**Soundness:** 2
**Presentation:** 2
**Contribution:** 2
**Rating:** 2
**Confidence:** 4

**Summary:**

This paper introduced the Removed-Masked Diffusion Model that enables a variable sequence length diffusion in Masked Diffusion Models. The core motivation of the paper is that the current discrete masked diffusion models (DMDs) incur significant computation in encoding each step sentence as a whole and output in a fixed length, which is not necessary and lacks flexibility. To address the challenge, the author redesigned the training and generation process by introducing a new [REMOVED] states while treating the original mask state as transient, so encoding only happens on unremoved tokens. In each generation, a lightweight gap counter estimator is applied to detect how many removed tokens should be considered in each step. While this method is well-motivated, the current experiment is weak and lacks empirical evidence to justify the advantages of Elastic-MDM over the recently developed DLM models.

**Strengths:**

1. The paper is well-motivated, with a clear definition of the current limitations of DMD.
2. The redesign of the state space is an interesting and practical idea, which naturally avoids unnecessary computations.

**Weaknesses:**

**Core:**
1. More baselines should be incorporated to justify the performance of this method, with both general task benchmarks and specialized tasks (i.e., math, code). The author should, for instance, evaluate Elastic-MDM on MMLU-pro for general tasks, GSM8K for math tasks, and MBPP/HumanEval for coding tasks.
2. The baselines compared are outdated. For now, there are many new DLM models (i.e., LLaDA, DREAM, etc.) that have been introduced in recent with better performance than the MDLM, yet these evaluations are largely missing. Without LLaDA/Dream, it’s hard to tell whether Elastic-MDM’s compute reductions translate into a better quality–latency frontier relative to today’s diffusion LMs. What was expected to see was whether Elastic-MDM can generate faster with similar or even better accuracy in comparison to these baselines (given a new training method proposed). 3. On the quality-speed tradeoff experiment against MDLM with LM1B, the evaluation focused on a short context (L = 128). Please include PPL-vs-latency at larger L (GPT2 at 512, Qwen3 at 1024, and larger context, $\geq$4096), given your OWT setting already covers L=1024.
4. The paper does not incorporate further details about how the lightweight gap count predictor was trained. A detailed, reproducible guidance with solid ablation is expected to address this concern.

**Minor:**

1. In line 707, there is a missing equation number “[??]”.

**Questions:**

(See weaknesses)

---

### Official Review · Reviewer_DHse · 2025-10-31

**Soundness:** 2
**Presentation:** 1
**Contribution:** 2
**Rating:** 2
**Confidence:** 4

**Summary:**

Current masked diffusion models are constrained to fixed-length sample generation, often resorting to tricks (e.g. padding) to deal with variable-length inputs. This paper addresses this issue by introducing a variable-length masked diffusion model based on token deletion/insertion in addition to masking/unmasking operations. This behavior is achieved by including a gap count prediction mechanism (distribution over number of tokens yet to be inserted at a given position) in addition to the canonical denoising loss.

The paper includes experiments on language modeling (measuring generative PPL and distribution of generated sequence lengths), a runtime analysis, and structured output generation.

**Strengths:**

- S1. The paper tackles an important problem: Variable length generation for discrete diffusion models and, specifically, for masked diffusion models (MDMs).
- S2. The key idea, a gap counting mechanism, is simple yet clever.

**Weaknesses:**

### W1. Execution
There is a significant lack of detail in many parts of the paper. I will list the most notable below.

#### W1.1. Computation cost and asymptotic order
The computation cost reported in Table 1 is stated without derivation. (Q1) How does one conclude the reported costs? For example, the cost of the exponential schedule assumes a constant $\sigma$, but surely larger $T$ call for smaller $\sigma$? A detailed derivation that clearly states all assumptions should be provided in the appendix.

The experimental analysis of this (Fig. 4) also raises many questions. For starters, the 3 plots should be combined into one for the sake of easy comparison (e.g. using one color per noise schedule and one line style per model type). Next, there seems to be a lot of noise in the data, with 150 steps (linear schedule) taking significantly more time than both 200 and 250 steps. This casts doubt on the accuracy of the reported numbers: (Q2) What sample size was used, and what are the confidence intervals on the reported numbers? The text further claims that the exponential schedule “yields asymptotically $O(L^2)$ behavior” (i.e. *constant* in $T$), but the reported numbers clearly show linear growth in the number of sampling steps with no signs of slowing down. (Q3) How do you explain this discrepancy?

#### W1.2. Length distribution
Much of the same issues are also found in Sec. 5.2 and Fig. 3. First, the plots are difficult to interpret (and the small font size and light colors are not helping); it would be better to plot the cumulative distribution of sequence lengths. (Q4) Like before, what is the sample size and confidence intervals associated with these numbers? Further, while the text claims that the length distribution of Elastic-MDM matches the data more closely than the baseline, from what I can tell, neither is particularly close. In fact, on OWT it appears that Elastic-MDM is even more concentrated around the maximum sequence length than the naive MDM baseline. (Q5) What is the quantitative overlap between the model and data sequence length distributions (e.g. measured via KL divergence or similar)?

#### W1.3. Perplexity vs. wall time
First, there are crucial details missing regarding the setup: (Q6) Which Qwen model was used for computing gen. PPL? (Q7) What are the denoising budgets used for the plotted points? (Q8) What is the sample size? Next, there are a number of oddities in the data. It seems like the walltime is non-monotonic in the number of denoising steps, with the first point in the line corresponding to Elastic-MDM (blue) taking less time than the second. Similarly, there exist points for MDM (red) that take the same amount of time, which is especially confusing given that MDM’s cost is linear and deterministic in the number of denoising steps. (Q9) What could be causing these discrepancies?

#### W1.4. Sloppy ELBO derivation
The derivation of the training loss (ELBO) provided in App. B.3 has a couple of issues. First, and more cosmetically, there is a reference error on L707 and a placeholder text that is suggesting the full derivation is still missing. Next, in Eqs. 21 ff., the expectation over $q(z_{t-1})$ should be conditioned on $z_t$ and $z_0$. There are also a lot of constants being pushed into $C$, making its effect on the likelihood bound potentially non-negligible. This raises concerns about the tightness of the proposed bound, and (Q10) calls for concrete numbers on the training/validation ELBO (including the constant terms) of Elastic-MDM vs. MDM. Related to this is the issue that many of the ELBOs reported in the paper (Eq. 3, 9, 10) are stated without constants $C$, which are crucial for the bounds to be correct, especially for Eq. 10 where the constant includes $O(T)$ individual terms. Finally, the proposed ELBO (Eq. 10) does not include any scaling terms, even though the first term (“denoiser loss”) arises from masked diffusion, where it is well-known that the cross-entropy term includes a scaling factor of $\frac{\alpha_{t-1} - \alpha_t}{1 - \alpha_t}$ (Eq. 3) and one should expect this term to also show up in Eq. 10, casting doubts on the tightness of the proposed ELBO.

#### W1.5 No results on training efficiency
While a core contribution of this paper compared to related work (specifically: Kim at al., 2025) is claimed to be to “address computation efficiency gained by omitting masked tokens”, the analysis is restricted to inference time, even though efficiency gains should also be observed at training time. (Q11) How does the training efficiency of Elastic-MDM compare to standard MDM?

#### W1.6. Lack of clarity regarding crucial design choices
- (Q12) In L294 it is stated that $t$ is drawn uniformly between $1, \dots, T$ (as is standard), but that $\alpha_t$ is also drawn uniformly from $[\beta, \omega]$, with $\beta$ and $\omega$ seemingly not depending on $t$. This is confusing: Is the noise level $\alpha_t$ indeed independent of $t$? If so, why worry about sampling $t$ at all? If not, what is the dependence of $\beta$ and $\omega$ on $t$?
- (Q13) In L301 it is stated that the proposed model predicts $x_{t-1}$ directly rather than $x_0$, but the loss from Eq. 10 suggests that the model is trained to maximize the likelihood of $x_0$. How do we reconcile this?
- (Q14) In Algo. 1, line 7, a binomial distribution is used to sample newly activated positions. Is this the correct choice? If so, what is the justification? And if not, could this potentially explain the mismatch in sequence length distribution observed between Elastic-MDM and the ground truth data?

### W2. No continuous-time formulation
The proposed ELBO (Eq. 10) does not include any sort of scaling factor for the inner terms, seemingly growing linearly in $T$, especially at initialization. As already touch upon in W1.4, this should be addressed by reporting training curves of the ELBO (including constant terms) for increasing $T$. If the proposed ELBO is indeed tight, then we should expect it to get tighter (and in practice lower) as $T$ increases. If the ELBO grows with $T$, then it is not tight and should not be used as a likelihood bound (even though it can still serve as a surrogate training objective).

---

In conclusion, the paper tackles an important problem and proposes a good idea (similar to Kim et al. (2025), which can be considered concurrent work), but the execution is poor.

**Questions:**

Questions
- Q1. How does one conclude the reported costs? For example, the cost of the exponential schedule assumes a constant $\sigma$, but surely larger $T$ call for smaller $\sigma$? (see W1.1)
- Q2. What sample size was used, and what are the confidence intervals on the reported numbers? (see W1.1)
- Q3. How do you explain this discrepancy? (see W1.1)
- Q4. What is the sample size and confidence intervals associated with these numbers? (see W1.2)
- Q5. What is the quantitative overlap between the model and data sequence length distributions (e.g. measured via KL divergence or similar)? (see W1.2)
- Q6. Which Qwen model was used for computing gen. PPL? (see W1.3)
- Q7. What are the denoising budgets used for the plotted points? (see W1.3)
- Q8. What is the sample size? (see W1.3)
- Q9. What could be causing this discrepancy? (see W1.3)
- Q10. What is the training/validation ELBO (including the constant terms) of Elastic-MDM vs. MDM? (see W1.4)
- Q11. How does the training efficiency of Elastic-MDM compare to standard MDM? (see W1.5)
- Q12-14: see W1.6
- Q13. How does the training ELBO behave for increasing $T$? (see W2)

Nits (not considered for final score):
- L80: Citation for MDLM should be Sahoo et al. (2024).
- Throughout (e.g. L90, L101): using `\cite{...}` instead of `\citep{...}` or vice versa
- L93: typo: “is excluded it”
- L142: Citation should be MDLM (Sahoo et al., 2024) not Sahoo et al. (2022).
- L240: Missing symbols in $l_t$ and $l_{t-1}$.
- L302-303: typo: “predicting $x_0$ directly from $x_0$”
- L667: “Details and edge cases (BOS/EOS) to be inserted.”
- L682: missing symbols inside $g_0()$
- L701: leading comma should be moved inside Eq. 17
- L704: What is “L_{RMDM}”?
- L707: “Full derivation to be inserted.”

---

### Official Review · Reviewer_CtQW · 2025-10-31

**Soundness:** 2
**Presentation:** 2
**Contribution:** 3
**Rating:** 4
**Confidence:** 4

**Summary:**

The paper proposes Elastic-MDM, a new masked diffusion model for text generation that (1) speeds up sampling by letting tokens that are already “finished” become a special [REMOVED] symbol and be completely dropped from future Transformer steps (so the model attends to fewer tokens over time), and (2) supports variable-length generation by predicting how many new tokens should be inserted in each gap between known tokens, so the sequence can grow or shrink instead of being fixed-length; the authors claim this yields lower sampling cost, comparable or better text quality, and better structured output (e.g. valid JSON) than a standard masked diffusion LM baseline. The paper is generally well motivated and the high-level ideas are understandable, but the presentation has several issues: inconsistent naming (Elastic-MDM vs RMDM), missing or placeholder equations and references (“Eq. ??”), incomplete derivations in the appendix, and figures/plots that illustrate trends without clear numeric axes or experimental details. These clarity and polish problems make it harder than it should be to reproduce the method or evaluate its claims in context with prior work.

**Strengths:**

S1. Originality.
The paper introduces a new discrete diffusion process where tokens can become an absorbing [REMOVED] state and are then physically pruned from the sequence, and where a learned gap-count head predicts how many new tokens to insert between surviving tokens. This jointly tackles efficiency and variable-length generation in a single generative mechanism, not just as ad hoc decoding tricks.

S2. Quality.
The approach targets two real limitations of masked diffusion LMs — slow sampling (recomputing attention over full-length sequences for many steps) and fixed-length outputs. The paper provides a complexity argument that pruning [REMOVED] tokens can reduce overall reverse-time attention cost toward 𝑂(𝐿^2)  instead of 𝑂(𝑇𝐿^2)  and shows empirical speed/quality trade-offs that are at least competitive with a masked diffusion baseline. It also shows a large jump in valid JSON generations (24.01% vs 3.58%), suggesting practical usefulness.

S3. Clarity.
The high-level story (“drop finished tokens to save compute; insert new masks to control length”) is clearly explained and aligned to the motivation. The paper gives intuitive descriptions of both forward and reverse processes and includes algorithm-style decoding steps. Plots convey the intended trends (latency vs perplexity, output length distribution).

S4. Significance.
If made fully rigorous and reproducible, this could influence how the community builds efficient non-autoregressive text generators: it offers a principled way to both cut sampling cost and generate flexible-length outputs, including structured outputs like JSON. This is directly relevant to current interest in diffusion-style LMs for controllable, tool-facing generation

**Weaknesses:**

W1. Incomplete theoretical derivation.
The paper claims a principled ELBO / NELBO-derived training loss that factorizes into (i) token denoising and (ii) gap-count prediction, and that avoids timestep weighting. However, the appendix is incomplete: there are placeholders and missing steps where the core derivation should be. The full checkable derivation of the reverse model
p_{theta,phi}(x_{t-1}, l_{t-1} | x_t, l_t),
how it factorizes, and the final claimed training loss
Loss = E_t[ - log p_theta(x_{t-1} | x_t, l_t)
            - log p_phi(l_{t-1} | x_t, l_t) ]
together with an explanation for why no timestep weighting term appears is missing. Without this, one of the main novelty claims (“we supervise the true reverse step, not a heuristic”) is not yet proven


W2. Underspecified decoding procedure.
The generation algorithm repeatedly (a) predicts how many tokens to insert in each gap, (b) inserts new [MASK] tokens, and (c) denoises them. Important details are missing:

How are positions assigned to newly inserted tokens? With RoPE, are you renumbering positions every step, interpolating, or reusing absolute indices?

How do you prevent runaway growth or oscillation (insert/delete/insert)? What concrete stopping rule and max length R_max were actually used in reported experiments? How exactly do you decide the final output length? These details are essential to know whether variable-length decoding is stable in practice, especially for long text


W3. Experimental evidence is underpowered.
The core empirical claims (“faster at similar quality”, “matches dataset length distribution”, “much better JSON validity”) are promising but not yet rigorous: Latency vs. quality plots are mostly qualitative curves without clear numeric axes (ms/sequence, tokens/sec), hardware/batch details, or error bars.  The length-distribution experiment shows histograms but no quantitative measure (e.g. KL divergence to the training length histogram, mean absolute length error) or controllability test (“generate ~50 tokens vs ~200 tokens”).
The JSON result (24.01% valid vs 3.58%) is compelling, but “valid” is not precisely defined and variance across samples is not reported.
Right now we get trend evidence, not reproducible numbers.


W4. Missing strong baselines.
The comparisons are mainly to a standard masked diffusion LM baseline that (1) keeps attending over the full canvas every step and (2) assumes fixed sequence length. There are two relevant baseline families that are not evaluated:

Insertion / edit / iterative refinement models that already support variable-length decoding.  Efficiency-focused diffusion LMs that cache or prune computation across steps (e.g. KV-cached or blockwise refinement methods). These baselines are cited in related work, but not reproduced. Without at least a small-scale direct comparison, it’s hard to tell if Elastic-MDM is actually advancing the frontier or just outperforming a baseline that was never optimized for these exact goals.


W5. Clarity / polish issues that affect credibility.
The draft still has issues like inconsistent naming (Elastic-MDM vs RMDM), dangling “Eq. ??” references, unfinished appendix sections, and notation drift (l_t vs L_t, gap indexing, etc.). Algorithm descriptions also leave BOS/EOS handling and stopping criteria implicit. These are all fixable, but in the current state they reduce confidence that all edge cases have actually been worked through.

**Questions:**

Q1. Can you provide the full derivation of your training objective?
Right now Appendix B says the loss directly supervises the reverse transition, and the final training loss is:

Loss = E_t[ - log p_theta(x_{t-1} | x_t, l_t)
            - log p_phi(l_{t-1} | x_t, l_t) ]

with no timestep weighting. Please provide a complete ELBO / NELBO derivation showing:

how you define the forward process over (x_t, l_t),

how you factorize the reverse model p_{theta,phi}(x_{t-1}, l_{t-1} | x_t, l_t),

why the timestep weighting terms that appear in standard masked diffusion disappear here.
This is central to the claimed novelty (“principled, not heuristic”), so I need to be able to verify it.


Q2. How exactly does your decoding algorithm remain stable when inserting tokens?
Your sampler repeatedly predicts gap lengths, inserts new [MASK] tokens, and denoises. Please clarify:

How are positional embeddings assigned to new tokens between two existing tokens? With RoPE, do you reindex positions every step, interpolate rotary angles, or something else?

What explicit stopping rule do you use at inference? Do you cap total active tokens (e.g. R_max), or do you terminate based only on the global diffusion step t→0?

How do you avoid oscillation (insert → remove → insert again)?
Right now, variable-length generation is a key claim, but the mechanics are only sketched.


Q3. Can you report quantitative efficiency numbers in a table, not just qualitative plots?
Please provide a table with: model size, hardware (GPU type/count), batch size, average generated length, number of denoising steps, wall-clock latency per generated sequence (ms/sequence or sequences/sec), and optionally tokens/sec. Include both Elastic-MDM and your strongest baseline.
The current plots show trends (“Elastic-MDM is faster for the same perplexity”) but do not expose exact magnitudes, axes, or variance, so it’s hard to judge how big the gain is in practice.



Q4. How is “generative perplexity” computed?
You say you evaluate sample quality by scoring generated text with a frozen LM evaluator. Please specify:

Which evaluator LM?

Are you using its native tokenizer/BPE on the generated text?

Are you sampling your model at a fixed temperature / nucleus setting?

How many samples and how long are they?

Do you average per-token negative log-likelihood over the full generated continuation?
Without these details, I can’t tell if perplexity differences are meaningful or just artifacts of evaluation setup.



Q5. Can you quantify the length-distribution match more rigorously?
You show that Elastic-MDM’s sampled sequence lengths roughly match the training length histogram, while the baseline collapses to a fixed max length. Could you report a metric (e.g. KL divergence between sampled length distribution and training length distribution, mean absolute error of length, etc.)? Also: can you steer the length? For example, if you bias the gap-count predictor toward “shorter” or “longer,” can you get ~50-token vs ~200-token outputs in a controlled way? This would strengthen the “controllable variable length” claim.




Q6. Why are certain baselines missing?
Your main comparisons are to standard masked diffusion LMs that are (a) fixed-length and (b) recompute attention over the full sequence each step. But you also cite work on insertion/edit-based generation (variable length) and KV/compute caching for diffusion (efficiency). Can you either (i) include at least one of these as a baseline, even on a small setting, or (ii) explain concretely why they are not directly comparable (e.g. need different training data, incompatible objective)?
Right now, this gap is one reason I’m only at “weak accept / borderline accept.”


Q7. How general is the method to very long sequences?
You argue that total attention cost across all denoising steps can approach O(L^2) instead of O(T * L^2) because inactive tokens get dropped early. Can you comment on behavior at much longer context lengths (e.g. 4K, 8K tokens)? Do you anticipate any failure modes, such as positional drift when repeatedly inserting between distant anchors, or degraded coherence because earlier spans were pruned too aggressively?

---

### Official Review · Reviewer_D8VH · 2025-11-02

**Soundness:** 3
**Presentation:** 3
**Contribution:** 4
**Rating:** 8
**Confidence:** 3

**Summary:**

Elastic-MDM tackles two core MDM issues—wasted compute and fixed-length outputs—by redefining the state space with an absorbing [REMOVED] token. In the forward process, tokens can transition to [REMOVED] and permanently drop out, so sequences shrink as noise increases. The reverse step pairs a Transformer denoiser (fills current masks) with a lightweight gap-count head (decides how many masks to re-insert between active tokens). Computation is applied only to active tokens, and a model-aligned ELBO yields a clean two-part loss (token reconstruction + gap counts) learned jointly per step.

On OBW and OpenWebText, quality (perplexity) matches a fixed-length MDM while running faster by skipping removed tokens; speedups grow with longer sequences/step counts, with reduced complexity (≈3× for linear schedules; O(L²) under exponential). Variable-length generation tracks the data’s length distribution, and on schema-conditioned JSON, valid outputs jump from 3.6% → 24% in a small-model setting. Net: a simple mask-as-transient redesign delivers a more efficient, length-flexible diffusion model with better fidelity on inherently variable-length, structured outputs.

**Strengths:**

The work tackles efficiency and variable-length generation in one framework. By augmenting the diffusion state with an absorbing removal state, the method cuts away both major bottlenecks of standard MDMs (excess compute on masked tokens and rigid output length)

The approach leads to speedups. The paper provides both complexity analysis and empirical timing to back this claim. For the same number of diffusion steps, Elastic-MDM achieved noticeably lower generation latency than the baseline, with larger gains for longer sequences. In the best case efficiency gains go from O(T·L^2) to O(L^2)

The introduction of the [REMOVED] token is conceptually simple yet grounded in a proper probabilistic diffusion framework. The forward process is defined with [REMOVED] as an absorbing state (once a token is removed, it stays removed) and this is mirrored by a well-specified reverse process with a joint distribution over tokens and sequence length. Further, the training objective is derived from first principles (the exact ELBO for the new state-space) rather than using ad-hoc loss weighting.
The wins are obvious in the structured space of JSON generation, it obeys schema constraints far more often than the baseline’s, specifically because it could terminate generation naturally at the appropriate length

**Weaknesses:**

Using an absorbing masking state in diffusion is not entirely new – e.g. DiffusionBERT and related works already treated the [MASK] as an absorbing end-state for corruption and skipping computation on masked tokens for efficiency has been explored (e.g. the concurrent EsoLM model excludes inactive tokens from attention, though without variable-length support. On the length side, several recent methods allow insertion/deletion or span diffusion to achieve variable-length generation

Elastic-MDM’s contribution is largely in marrying these existing concepts (efficient masking and length flexibility) - but i believe the performance and efficiency gains deserve publicaiton - the combination of approaches has worked better than one would necessarily expect that that's notworthy.

The baselines could be stronger - MDM and SEED only. Maybe coukd compare with DDPD, FlexMDM or DDOT which are not used as comparisons, just mentioned.

There are extra moving parts that need independent tuning and ad hoc modeling choices ( the forward process only removes, but never extends), so there's still a maximum, maximum length. Does the gap predictor need to be called at every diffusion step ? What about predicting insertion position explicitly ?

**Questions:**

N/A

---

### Meta-Review · Area_Chair_5v3X · 2026-01-08

**Summary:**

The reviewers raised concerns about the incomplete theoretical foundation, weak baselines, algorithmic ambiguity, and questionable empirical evidence; however, the authors did not offer any response. The reviews were mixed: three reviewers gave negative scores (including two reject), while only one was satisfied with the results.

**Reviewer Concerns:**

As the authors did not provide any response during the rebuttal period, the reviewers’ concerns remain entirely unaddressed

**Reviewer Scores:**

As the authors did not provide any response during the rebuttal period, the reviewers are expected to retain their original scores.

---

### Decision · Program_Chairs · 2026-01-26

Reject